# Sex and Age-Related Differences in Neuroinflammation and Apoptosis in *Balb/c* Mice Retina Involve Resolvin D1

**DOI:** 10.3390/ijms22126280

**Published:** 2021-06-11

**Authors:** Maria Consiglia Trotta, Sami Gharbia, Hildegard Herman, Bianca Mladin, Andrei Hermenean, Cornel Balta, Coralia Cotoraci, Victor Eduard Peteu, Carlo Gesualdo, Francesco Petrillo, Marilena Galdiero, Roberto Alfano, Mihaela Gherghiceanu, Michele D’Amico, Settimio Rossi, Anca Hermenean

**Affiliations:** 1Section of Pharmacology, Department of Experimental Medicine, University of Campania “Luigi Vanvitelli”, via Santa Maria di Costantinopoli 16, 80138 Naples, Italy; mariaconsiglia.trotta2@unicampania.it (M.C.T.); marilena.galdiero@unicampania.it (M.G.); 2“Aurel Ardelean” Institute of Life Sciences, Vasile Goldis Western University of Arad, 86 Revolutiei Av., 310144 Arad, Romania; gharbia.sami@uvvg.ro (S.G.); herman.hildegard@uvvg.ro (H.H.); mladin.bianca@uvvg.ro (B.M.); balta.cornel@uvvg.ro (C.B.); hermenean.anca@uvvg.ro (A.H.); 3Faculty of Medicine, Carol Davila University of Medicine and Pharmacy, 8 Eroii Sanitari Av., 050474 Bucharest, Romania; Andrei.hermenean@gmail.com (A.H.); mihaela.gherghiceanu@ivb.ro (M.G.); 4Faculty of Medicine, Vasile Goldis Western University of Arad, 86 Revolutiei Av., 310144 Arad, Romania; cotoraci.coralia@uvvg.ro; 5Victor Babes National Institute of Pathology, 99-101 Splaiul Independentei Av., 050096 Bucharest, Romania; peteuvictoreduard@gmail.com; 6Eye Clinic, Multidisciplinary Department of Medical, Surgical and Dental Sciences, University of Campania “Luigi Vanvitelli”, Via Luigi De Crecchio 6, 80138 Naples, Italy; carlo.gesualdo@unicampania.it (C.G.); settimio.rossi@unicampania.it (S.R.); 7Department of Ophthalmology, University of Catania, P.zza Università 2, 95131 Catania, Italy; francescopetrillo09@gmail.com; 8Department of Advanced Medical and Surgical Sciences “DAMSS”, University of Campania “Luigi Vanvitelli”, P.zza L. Miraglia 2, 80138 Naples, Italy; roberto.alfano@unicampania.it

**Keywords:** retina, aging, neuroprotection, apoptosis

## Abstract

(1) Background: The pro-resolving lipid mediator Resolvin D1 (RvD1) has already shown protective effects in animal models of diabetic retinopathy. This study aimed to investigate the retinal levels of RvD1 in aged (24 months) and younger (3 months) *Balb/c* mice, along with the activation of macro- and microglia, apoptosis, and neuroinflammation. (2) Methods: Retinas from male and female mice were used for immunohistochemistry, immunofluorescence, transmission electron microscopy, Western blotting, and enzyme-linked immunosorbent assays. (3) Results: Endogenous retinal levels of RvD1 were reduced in aged mice. While RvD1 levels were similar in younger males and females, they were markedly decreased in aged males but less reduced in aged females. Both aged males and females showed a significant increase in retinal microglia activation compared to younger mice, with a more marked reactivity in aged males than in aged females. The same trend was shown by astrocyte activation, neuroinflammation, apoptosis, and nitrosative stress, in line with the microglia and Müller cell hypertrophy evidenced in aged retinas by electron microscopy. (4) Conclusions: Aged mice had sex-related differences in neuroinflammation and apoptosis and low retinal levels of endogenous RvD1.

## 1. Introduction

The decrease in the physiological functions of the aged organs can be influenced by genetic and environmental factors [1]. Age-related immune dysfunctions leading to chronic inflammation is a major risk factor for the incidence and prevalence of age-related diseases, including neurodegenerative diseases [2]. Retinal neurodegenerative diseases are associated with neuroinflammation, chronic activation and proliferation of the microglia, alongside with neuronal and glial cell death [3], that leads to morphological alterations and visual impairment [4].

Clinical studies have indicated that eye disorders, such as cataract, glaucoma, and age-related macular degeneration (AMD), are associated with sex and have an increased incidence with age [5,6,7].

In this context, our previous work showed that the aged retina is more sensitive to the damage in male mice than in female mice [8]. This is because, in the retina of male mice, there is a greater dysregulation of some age-related microRNAs (miRNAs) that link to oxidative stress response and neurodegeneration. In fact, the thickening of the retina and the integrity of the Bruch’s membrane were correlated with the dysregulation of miR-27a-3p, miR-20a-5p, miR-20b-5p, and miR-27b-3p in physiologically aged male mice compared to physiologically aged female mice [8].

Since the ocular expression of some of these miRNAs is under control of the pro-resolving lipid mediator Resolvin D1 (RvD1) in male murine models of retina degeneration [9,10,11,12] and RvD1 itself governs neurodegenerative disorders with protective roles, we thought about its possible involvement in the sex and age differences in aged retinas compared to younger retinas. Therefore, apoptosis, neuroinflammation, and activated macro and microglia together with the retinal levels of RvD1 are studied here in the aged and younger retina of male and female *Balb/c* mice.

## 2. Results

### 2.1. RvD1 Levels in Aged Retina

Control male (CM) and female (CF) mice exhibited similar levels of retinal RvD1 (CM: 80 ± 9 IU/mL; CF: 70 ± 11 IU/mL). On the contrary, RvD1 was markedly decreased in the retina of aged male (AM) mice (20 ± 5 IU/mL; *p* < 0.01 vs. CM). This trend was similar in aged females (AF) (40 ± 8 IU/mL; *p* < 0.05 vs. CF), although these showed significantly higher retinal RvD1 levels compared to aged males (*p* < 0.05) (Figure 1).

### 2.2. Microglial Activation

Ionized calcium-binding adapter molecule 1 (Iba-1), as a marker of microglial activation, was inversely correlated with the levels of RvD1 measured in the retina homogenates (r = −0.88; *p* < 0.01) (Table 1). Increased Iba-1 immunoreactivity (as a marker of microglial activation) was detected in aged retina compared to control retina. This was distributed in outer retina (starting with photoreceptor outer segment) and inner retina evenly throughout the nerve fiber layer and the ganglion cell layer (Figure 2A). The retinas of both aged male and female mice showed a significant increase in Iba-1 (AM: 79 ± 8%, *p* < 0.01 vs. CM; AF: 45 ± 10%, *p* < 0.05 vs. CF) compared to the retinas of young same-sex mice (CM: 20 ± 2%; CF: 25 ± 3%) (Figure 2A). However, it was more markedly significant for the retinas of aged male mice than for the retinas of aged female mice (*p* < 0.05 vs. AM) (Figure 2A). These results were confirmed by Iba-1 protein levels’ detection by Western Blotting analysis (Figure 2B). Microglial hypertrophy was evidenced in the extended damaged inner nuclear layer (INL) areas in aged retina of both sexes (Figure 2C).

### 2.3. Müller Cell Activation

The regression with the RvD1 levels showed a significant negative correlation between RvD1 and glial fibrillary acidic protein (GFAP), a marker of astrocytes activation (r = −0.86; *p* < 0.01) (Table 1). Particularly, GFAP-positive cells were marked in aged retina (AM: 58 ± 5%, *p* < 0.01 vs. CM; AF: 37 ± 4%, *p* < 0.05 vs. CF) (Figure 3A). These were localized in the inner and outer retinal layers and were much more intense in the retinas of male mice if compared to the retinas of female mice (*p* < 0.01 vs. AM) (Figure 3A). Accordingly, GFAP protein levels detected by Western Blotting showed the same trend between the control and aged retina, particularly between aged male and female mice (*p* < 0.01 vs. AM) (Figure 3B). Moreover, electron microscopy evidenced Müller cell hypertrophy and hyperplasia on aged retina of both sex (Figure 3C). Particularly, a normal aspect of the Müller cells processes between bipolar cells was present in the inner nuclear layer (INL) of the controls, while an extended network of cytoplasmic process of Müller cells was shown by both aged retinas (Figure 3C).

### 2.4. Neuroinflammation

Aged retinas showed increased markers of inflammation such as nuclear factor kappa-light-chain-enhancer of activated B cells (NF-kB) (Figure 4) and Tumor Necrosis Factor alpha (TNF-α) (Figure 5) detected by immunohistochemistry compared to controls. Particularly, NF-kB positive staining in aged males showed a % of 87 ± 11 (*p* < 0.01 vs. CM) and of 64 ± 4 in aged females (*p* < 0.05 vs. CF). These latter were significantly different from aged males (*p* < 0.05 vs. AM) (Figure 4A,B). Similarly, TNF-α positive cells were increased in aged retinas (AM: 80 ± 8%, *p* < 0.01 vs CM; AF: 59 ± 5%, *p* < 0.01 vs. CF) (Figure 5A,B), significantly differing by sex (AF *p* < 0.05 vs. AM) (Figure 5A). ELISA of both NF-kB (Figure 4C) and TNF-α (Figure 5C) protein levels were in line with this trend. Both NF-kB and TNF-α were inversely correlated with RvD1 levels (r = −0.93 and r = −0.94, respectively, both *p* < 0.01) (Table 1).

### 2.5. Apoptosis

Retinal cell apoptosis is illustrated in Figure 6, showing caspase-3 positive cells in retinal layers. Only cells from aged retinas showed a high presence of apoptosis as caspase-3 (AM: 88 ± 12%, *p* <0.01 vs. CM; AF: 57 ± 10%, *p* < 0.05 vs. CF) (Figure 6A,B). This marker intensely presents in aged retinas of male mice (Figure 6Aa,b) while weaklier in retinas of female mice (*p* < 0.05 vs. AN) (Figure 6Ac,d). These results were confirmed by caspase-3 protein levels detection by ELISA (Figure 6C). Moreover, correlation analysis showed a negative association between RvD1 and caspase-3 levels (r = −0.82, *p* < 0.01) (Table 1).

### 2.6. Nitrotyrosine Measurement

The measure of the 3-nitrotyrosine levels as a marker of peroxynitrite formation in the retina homogenates was high in the aged retina of both sexes (AM: 95 ± 8 ng/mL, *p* < 0.01 vs. CM; AF: 74 ±7 ng/mL, *p* < 0.05 vs. CF) (Figure 7). This had the maximal values in retinas extracted from eyes of male mice being significantly different from retinas of female mice aged equally (*p* < 0.05 vs. AM) (Figure 7). Additionally, 3-nitrotyrosine levels were negatively correlated with RvD1 levels (r = −0.87, *p* < 0.01) (Figure 7).

## 3. Discussion

RvD1 is a lipid derived from docosahexaenoic acid metabolism together with protectins and maresins [13]. It binds its own specific receptor called formyl peptide receptor 2 (FPR2) and is involved in the genesis/resolution of several inflammatory pathologies [11]. The deficiency of this lipid has been linked to the onset of degenerative diseases typical of the CNS, especially if an oxidative-inflammatory insult causes the degeneration of the cells involved [14,15,16]. Accordingly, in previous papers done by this group, RvD1 showed a protective effect when applied exogenously to murine uveitic eyes and degenerating photoreceptors in vitro [9,10,11,12]. Here, we further contribute to these data by defining for the first time a key involvement of RvD1 in apoptosis and neuroinflammation occurring in the physiologically aged retina. Two novelties are evidenced: (i) the mediator RvD1 decreased in the 24-months-old retina, and (ii) it had different levels in the retina of the male mice compared to the retina of the female mice. Phenomena associated with greater damage to specific segments of the retina in males than in females (e.g., retinal pigment epithelium and Bruch’s membrane, outer and inner layers) [8]. From the mechanistic point of view, the decrement of RvD1 was paralleled by microglia activation together with gliosis and increased apoptotic and nitrosative response into the retina.

Retinal damage is due to local immune-inflammatory processes affecting the RPE, photoreceptors layer, ganglion, and nerve fiber layers and inner nuclear layer (INL) [17], along with glial cells activation [18,19]. There are two glial retinal types with specific morphology, physiology, and antigenicity: macroglia (Müller cells and astrocytes) and microglia [20]. Activated Müller cells can induce gliosis and contribute to neuron death by secretion of proinflammatory factors, i.e., TNFα, monocytic chemotactic type 1 protein (MCP1), interleukins, interferon, and nitric oxide (NO), leading to free radicals’ release and protein nitrosylation, with neuronal toxicity effects [21,22,23]. In line with this, retinal reactive gliosis (astrocytosis) increased since GFAP labelling was pronounced in aged retina compared to young retina, particularly in male retina compared to female retina. This was associated with microglia activation in aged mice compared to adult mice, specifically much more intensely in the retina of aged male mice than aged female mice. This latter example underlines a lesser extent of neuroinflammatory damage. The microglia activation process is a complex phenomenon, characterized by the acquisition of different functional phenotypes, schematically represented by the M1 and M2 phenotypes, associated respectively with neuro-toxic and neuroprotective functions. Accordingly, increased Iba-1 indicated that the M1 phenotype was present in aged retina.

Activated macro- and microglia cause neuroinflammation and increase retinal apoptosis [24], the final stage of cellular damage aimed at the removal of undesired cells [25]. However, dysregulation of the apoptotic mechanisms (e.g., persistent immune-inflammation, mitochondrial damage, ROS generation, and epigenetic alterations) may be disadvantageous since it may lead to increased cell loss, tissue dysfunction, and exacerbated postmitotic cell (neurons)-associated pathological conditions [25]. Retina is one tissue that is highly exposed to cellular damage because of the prolonged exposition to damaging factors such as light, microbes, and chemicals [26,27] that may cause apoptosis and cell death in the long run, as of the example of RPE cells [28]. Here, apoptosis increased, with aged male retina showing more apoptosis than female retina. Interestingly, RvD1 was much lower in the male retina than in the female retina.

Another aspect of aged retina is the presence of nitrosative stress, which exerts damaging effects [29]. In the mammalian retina, NO has been detected in amacrine cells, bipolar, and ganglion cells in the inner retina, whereas interplexiform cells, bipolar cells, and horizontal cells are sources of NO for the outer retina [30]. However, an overproduction of NO generates NOx production, such as peroxynitrite, which deranges retinal structure by reacting with several biomolecules and potentially leads to cell death [29,31]. Accordingly, here, we recorded higher levels of nitrotyrosine (index of peroxynitrites) in aged retina with respect to younger retina. In particular, the levels of nitrotyrosine were lower in the retina of aged female mice than in the retina of aged male mice.

In conclusion, the retinal RvD1 levels were decreased in aged mice when compared to younger mice, and the decrease was markedly larger in males. Several aspects of the aged retina (the astrocyte activation, neuroinflammation, apoptosis and nitrosative stress, being in line with Müller cell hypertrophy) were paralleled by changes of RvD1 levels. Considering that retinal aging is a progressive process, more in-depth studies should progressively monitor these alterations, even in different mouse strains.

## 4. Materials and Methods

### 4.1. Animals

Experimental procedures were conducted according to the guidelines of the Declaration of Helsinki, in compliance with European and national guidelines for research on laboratory animals and had the ethical approval from the Vasile Goldis Western University of Arad (Approval no.135/2019).

Male and female *Balb/c* mice aged 3 months (control groups) and 24 months (aged groups; approximately 75–85 years for humans) [8,32], respectively (*n* = 10/each group/sex), were used. Young females were under physiological and regular estrus cycle, while aged females were under naturally occurring physiological decline of estrus without any manipulation [8,33]. All were housed in IVC cages, in standard temperature and humidity conditions, with ad libitum access to food and water. Lighting was regulated on a 12-h light/dark cycle. Particularly, to minimize the negative effects of standard vivarium lighting on the aged retina, an illuminance level of 39 ± 7 lux was used [8,33]. This was even lower than the room light recommended for animals susceptible to phototoxic retinopathy (between 130 and 325 lux) by the National Research Council (US) Committee for the Update of the Guide for the Care and Use of Laboratory Animals [34].

Once the experimental setting has been prepared, each mouse under anaesthesia had systemic perfusion via the left ventricle with 100 mL of 0.1 M ice-cold phosphate-buffered saline (PBS) + heparin (5000 IU/mL, final concentration of 0.1% *v*/*v*) [35]. At the end of perfusion, one eye for biochemical assays was excised. In a next step, animals’ perfusion was continued with 100 mL more of freshly prepared 4% paraformaldehyde (PFA) in PBS for the collection of the remaining eyes and investigations detailed below.

### 4.2. Immunohistochemistry

Immunohistochemistry was done on 5 µm paraffin-embedded eye sections, previously deparafinized and rehydrated using a standard technique. Primary antibodies diluted 1:200: mouse monoclonal caspase-3 (sc-271759; Santa Cruz Biotechnology; Dallas, TX, USA), rabbit polyclonal NF-kB (sc-109; Santa Cruz Biotechnology; Dallas, TX, USA), mouse monoclonal TNF-α (ab 1793; Abcam PLC., Cambridge, UK) were incubated overnight la 4 °C.

Novocastra Peroxidase/DAB kit (Leica Biosystems, Nussloch, Germany) was used to detect immunoreactions, according to the manufacturer’s instructions. The substitution of primary antibodies with irrelevant immunoglobulins of matched isotype was used to stain negative control sections and all were analysed under bright-field microscopy.

### 4.3. Immunofluorescence

GFAP levels were assessed by using a rabbit polyclonal anti-GFAP antibody (ab7260; Abcam PLC., Cambridge, UK) and AlexaFluor 594 labeled goat anti-rabbit IgG secondary antibody (A 11037; Thermo Fisher Scientific Inc., Rockford, IL, USA). Goat polyclonal Iba-1 (ab-5076; Abcam PLC., Cambridge, UK) and donkey anti-goat AlexaFluor 594 (a-11058; Invitrogen, Waltham, MA, USA) were used as primary antibody and secondary antibody, respectively. Bond Dewax Solution (Leica Biosystems Inc., Buffalo Grove, IL, United States) was used to deparaffinate the eye sections. They were rehydrated in alcoholic solutions. Epitope Retrieval Solution (Leica Biosystems Inc., Buffalo Grove, IL, United States) was used for antigen retrieval at 95 °C for 10 min, followed by blocking with 2% BSA in PBS. The primary antibody was applied in a dilution of 1:1000 in primary antibody diluting buffer (Bio-Optica, Milano, Italy) for 2 h at 4 °C. The slides were washed in PBS and incubated with the secondary antibody, diluted to 1:500 in PBS, for 2 h at room temperature in the dark. After a further 3 washing steps with PBS, nucleus counterstaining was performed with 1 μg/mL DAPI (Sigma-Aldrich, St Louis, MO, USA). CC/Mount aqueous mounting medium (Sigma-Aldrich, St Louis, MO, USA) was used to mount the stained slides. They were examined with a Leica SP5 confocal laser scanning microscope.

### 4.4. Transmission Electron Microscopy

The eye samples were prefixed in 2.7% glutaraldehyde solution (Sigma-Aldrich, St Louis, MO, USA) in 0.1 M phosphate buffer; then, washed in 0.15 M phosphate buffer (pH 7.2) and post-fixed in 2% osmic acid solution (Sigma-Aldrich, St Louis, MO, USA) in 0.15 M phosphate buffer. Acetone was used for dehydration, followed by inclusion in the epoxy embedding resin Epon 812. Ultrathin sections of 70 nm sections were cut on Leica EM UC7 ultramicrotome (Leica Microsystems GmbH, Wetzlar, Germany) and doubly contrasted with solutions of uranyl acetate and lead citrate and analyzed with TEM (Morgagni 268, FEI, Eindhoven, Netherlands) at 80 kV. Data acquisition was performed with a MegaView III CCD using iTEM SIS software (Olympus Soft Imaging Software, Munster, Germany).

### 4.5. Western Blotting

Retinas were first dissected as previously described [36], then homogenized in RIPA lysis buffer (R0278; Sigma-Aldrich, Milan, Italy) containing a protease inhibitor cocktail (11873580001; Roche, Monza, Italy). After a centrifugation at 13,000× *g* for 10 min at 4 °C and the subsequent separation of nucleic acids from protein supernatants, total protein concentration was assessed by using Bio-Rad protein assay protocol (500-0006, Bio-Rad Laboratories; Segrate, Italy). Western blotting assay was performed as previously described [37]. Briefly, protein samples were separated by SDS-electrophoresis in an 8% polyacrylamide gel and then electrotransferred onto PVDF membranes (Thermo Fisher Scientific Inc., Rockford, IL, USA). These were blocked for 1 h at room temperature with 5% non-fat dry milk (EMR180500; Euroclone SpA, Milan, Italy) with PBS-T (PBS-0.05% Tween 20) (P1379-500ML; Sigma-Aldrich, Milan, Italy) before the incubation overnight at 4 °C with anti-GFAP (0.1 µg/ mL; ab53554; Abcam PLC., Cambridge, UK), anti-Iba-1 (2 µg/mL; 5076; Abcam; PLC., Cambridge, UK), and anti-actin (C-2) (1:200 in blocking solution; sc-8432, Santa Cruz Biotech, CA, USA) primary antibodies dissolved in PBS-0.05% Tween 20. Anti-goat (sc-2020; Santa Cruz Biotech, CA, USA) and anti-mouse (sc-2005; Santa Cruz Biotech, CA, USA) horseradish peroxidase-conjugated secondary antibodies (1:2000 in PBS-T; Santa Cruz Biotech, CA, USA) were incubated at 1 h at room temperature to detect immunoreactive signals. These were visualized with an ECL system (Amersham Pharmacia, Uppsala, Sweden), quantized by ChemiDoc-It 5000 by using VisionWorks Life Science Image Acquisition and Analysis software (UVP, Upland, CA, USA), normalized with actin levels, and expressed as densitometric units (DU).

### 4.6. ELISAs

Levels of RvD1 (MBS2600566; MyBiosource; San Diego, CA, USA), nitrotyrosine (as marker of peroxynitrite formation) (ab116691; Abcam PLC., Cambridge, UK), NF-kB p65 (ab176648; Abcam PLC., Cambridge, UK), TNF-α (MBS825075; MyBiosource; San Diego, CA, USA) and caspase-3 (E4591-100; BioVision; Milpitas, CA, USA) were determined in retinal tissues by commercial ELISA tests, following the manufacturer’s protocols.

### 4.7. Statistical Analysis

Results are expressed as the mean ± standard error of the mean (S.E.M.). One-way ANOVA followed by Tukey’s multiple comparisons test was used to assess statistical significance. The strength of the association between pairs of variables was evaluated by Pearson correlation analysis. GraphPad Prism (6.0 GraphPad Software, La Jolla, CA, United States) was used to carry out statistical analysis, by considering differences for *p* values < 0.05 statistically significant.

## Figures and Tables

**Figure 1 ijms-22-06280-f001:**
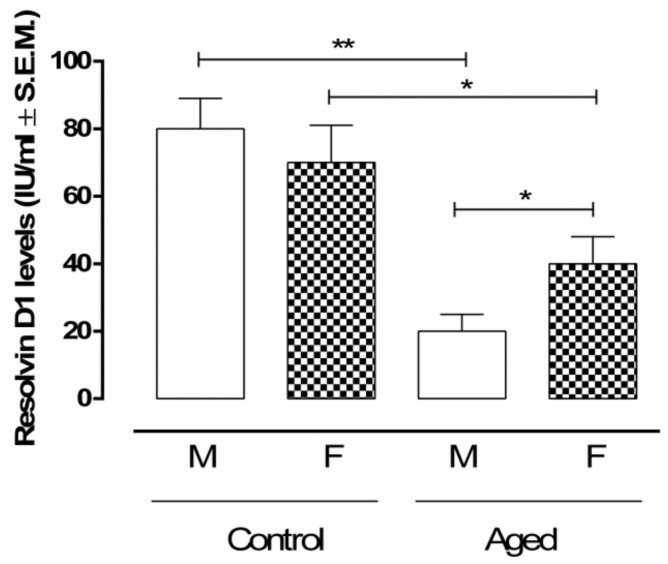
RvD1 retinal levels. ELISA determination of RvD1 in aged and younger (control) retina. Data are expressed as mean ± S.E.M. of N = 10 retinas per group. M: males; F: females; IU: International Units; ** *p* < 0.01; * *p* < 0.05 vs. control.

**Figure 2 ijms-22-06280-f002:**
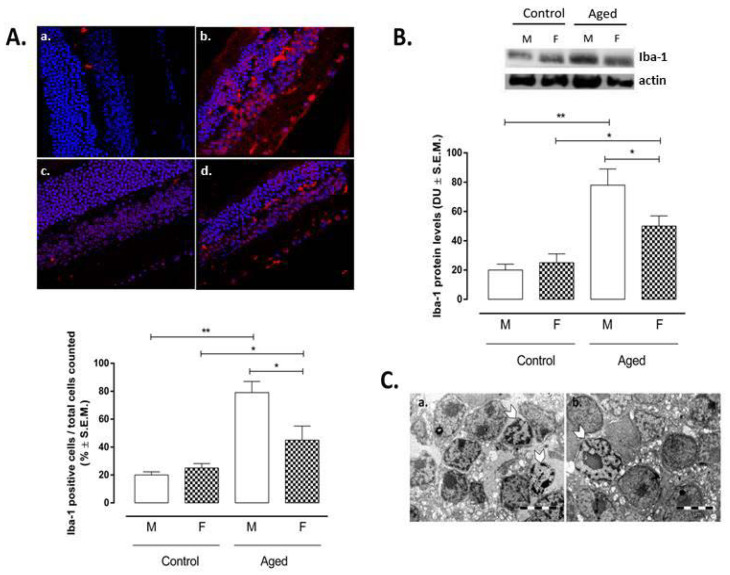
Microglial activation in aged and younger (control) retina. (**A**) Immunofluorescence staining in control and aged retina. Immunofluorescence images and data (% ± S.E.M.) are representative of 10 observations for each individual/group; Iba-1 immunolabeling of the retina indicates less reactive microglia in the controls (a. male; c. female); aged samples (b. male; d. female) contain activated macroglia cells which were positive for Iba-1; ** *p* < 0.01; * *p* < 0.05 vs. control; magnification 63×; (**B**) Western blotting results are expressed as the mean ± S.E.M. of *n* = 10 retinas per group. M: males; F: females; DU: densitometric units; ** *p* < 0.01; * *p* < 0.05 vs. control; (**C**) electron microscopy evidenced microglial hypertrophy (arrowhead) in the retinal inner nuclear layer (INL) of aged male (a) and aged female (b); bar 5 µm.

**Figure 3 ijms-22-06280-f003:**
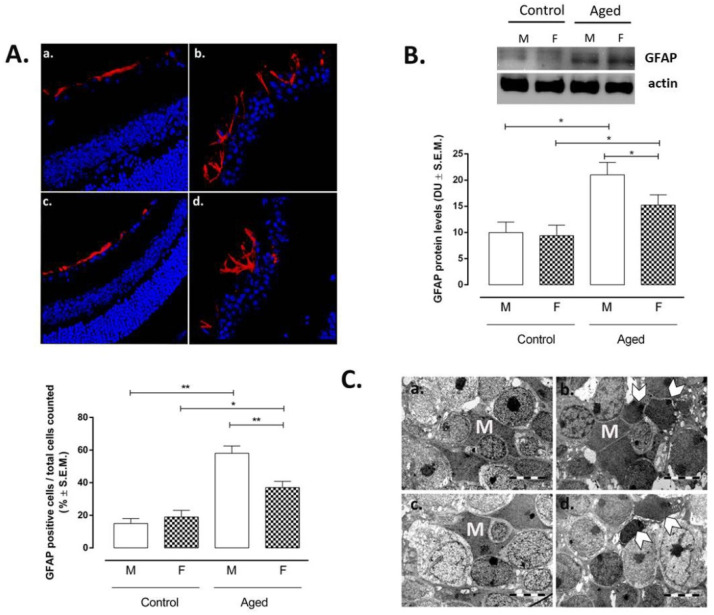
Müller cell activation in aged retina. (**A**) Immunofluorescence staining in control and aged retina. Immunofluorescence images and data (% ± S.E.M.) are representative of 10 observations for each individual/group; significantly increased GFAP immunoreactivity was noted in aged retinas (b. male; d. female), compared with the normal controls (a. male; c. female); ** *p* < 0.01; * *p* < 0.05 vs. control; magnification 63×; (**B**) Western blotting results are expressed as the mean ± S.E.M. of *n* = 10 retinas per group. M: males; F: females; DU: densitometric units; ** *p* < 0.01; * *p* < 0.05 vs. control; (**C**) electron microscopy evidenced a normal aspect of the Müller cells processes between bipolar cells in the inner nuclear layer (INL) of the controls (a. control male, c. control female) and extended network of cytoplasmic process of Müller cells (M); in both aged retinas (b. aged male; d. aged female); hypertrophied Müller cells (arrows); bar 5 µm.

**Figure 4 ijms-22-06280-f004:**
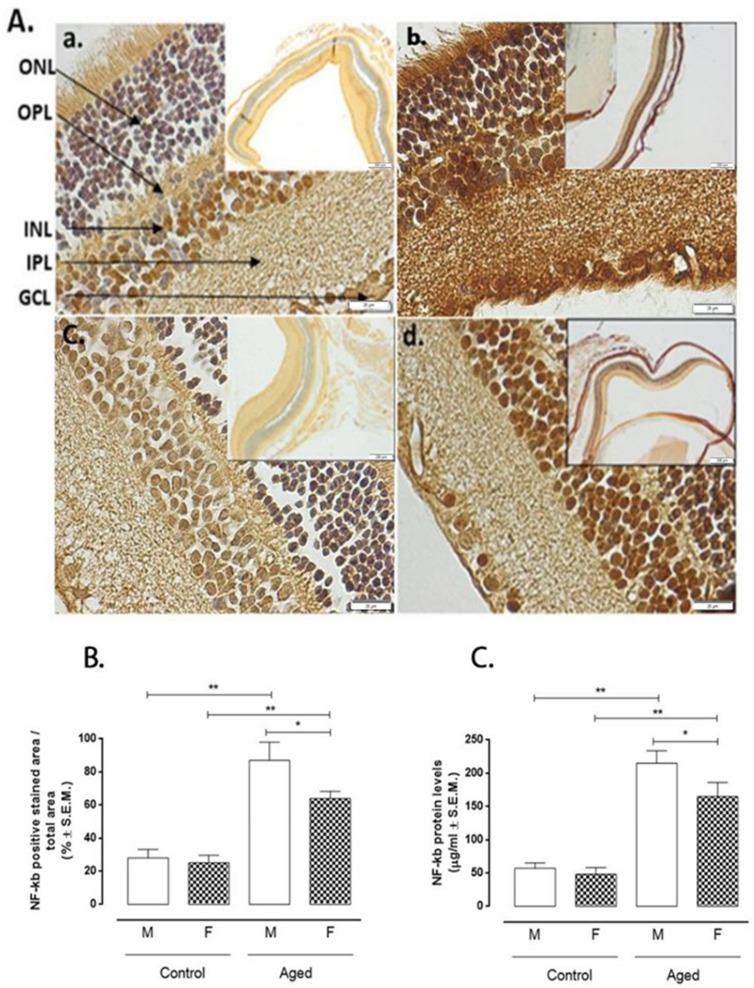
NF-kB expression in aged and younger (control) retina. (**A**) Representative immunohistochemistry (a. control male; b. aged male; c. control female; d. aged female) and (**B**) quantification of immunopositive areas in control and aged retina (M: males and F: females); the results calculated as the % ± S.E.M. are considered statistically significant when * *p* < 0.05; ** *p* < 0.01 vs. control; *n* = 10 observations for each individual/group; ONL: outer nuclear layer; OPL: outer plexiform layer; INL: inner nuclear layer; IPL: inner plexiform layer; GCL: ganglion cell layer; Panel A: bar = 20 µm. The frames in the panel A are given on the right in a lower magnification bar = 200 µm; (**C**) enzyme-linked immunosorbent assay (ELISA) of NF-kB protein levels in the control and aged retina; results are expressed as the mean ± S.E.M. of *n* = 10 retinas per group. DU: densitometric units; ** *p* < 0.01; * *p* < 0.05 vs. control.

**Figure 5 ijms-22-06280-f005:**
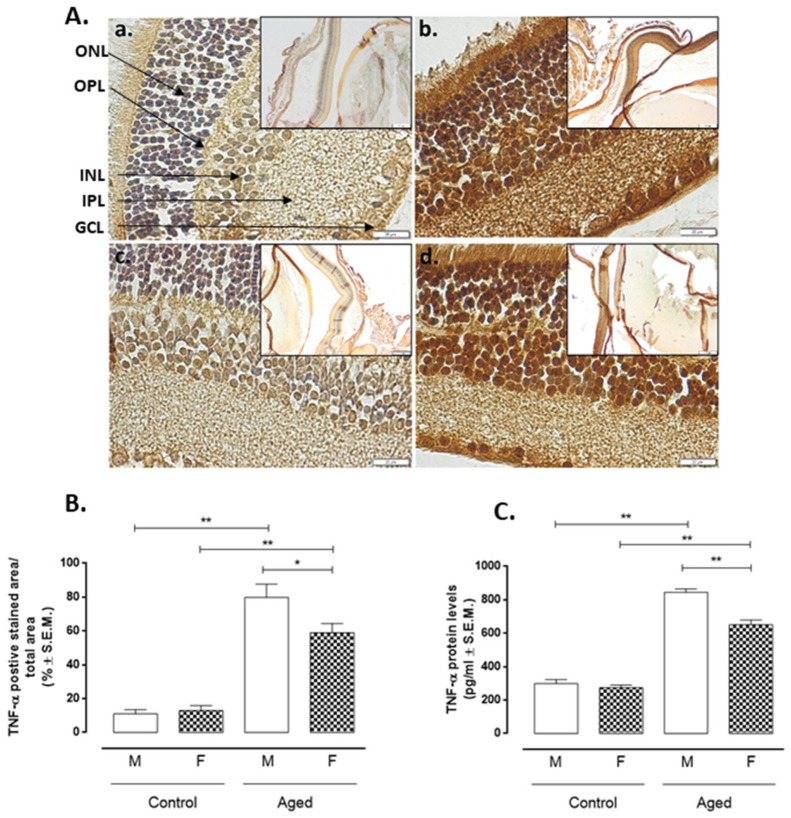
TNF-α levels in aged and younger (control) retina. (**A**) Representative immunohistochemistry of TNF-α (a. control male; b. aged male; c. control female; d. aged female) and (**B**) quantification of immunopositive areas in control and aged retina (M: males and F: females); the results calculated as the % ± S.E.M. are considered statistically significant when * *p* < 0.05; ** *p* < 0.01 vs. control; *n* = 10 observations for each individual/group; ONL: outer nuclear layer; OPL: outer plexiform layer; INL: inner nuclear layer; IPL: inner plexiform layer; GCL: ganglion cell layer; Panel A: bar = 20 µm. The frames in the panel A are given on the right in a lower magnification bar = 200 µm; (**C**) ELISA of TNF-α protein levels in the control and aged retina; results are expressed as the mean ± S.E.M. of *n* = 10 retinas per group. DU: densitometric units; ** *p* < 0.01; * *p* < 0.05 vs. control.

**Figure 6 ijms-22-06280-f006:**
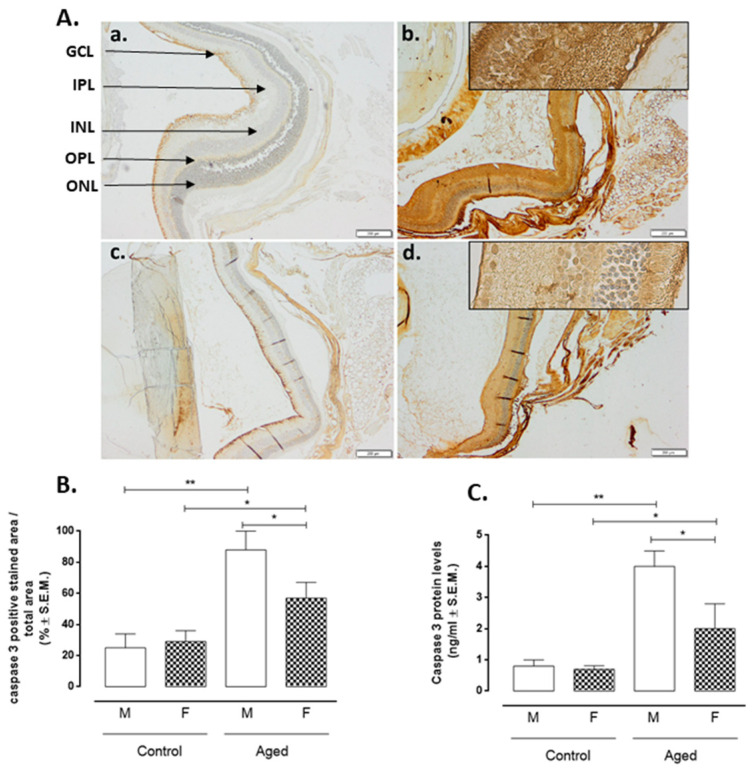
Caspase-3 expression levels in aged and younger (control) retina. (**A**) Representative immunohistochemistry of caspase-3 (a. control male; b. aged male; c. control female; d. aged female) and (**B**) quantification of immunopositive area in control and aged retina (M: males and F: females); the results calculated as the % ± S.E.M. are considered statistically significant when * *p* < 0.05; ** *p* < 0.01 vs. control; *n* = 10 observations for each individual/group; ONL: outer nuclear layer; OPL: outer plexiform layer; INL: inner nuclear layer; IPL: inner plexiform layer; GCL: ganglion cell layer; Panel A: bar = 200 µm; the frames in the panel A are given on the right in a higher magnification bar = 20 µm; (**C**) ELISA of caspase-3 protein levels in control and aged retina; results are expressed as the mean ± S.E.M. of *n* = 10 retinas per group. DU: densitometric units; ** *p* < 0.01; * *p* < 0.05 vs. control.

**Figure 7 ijms-22-06280-f007:**
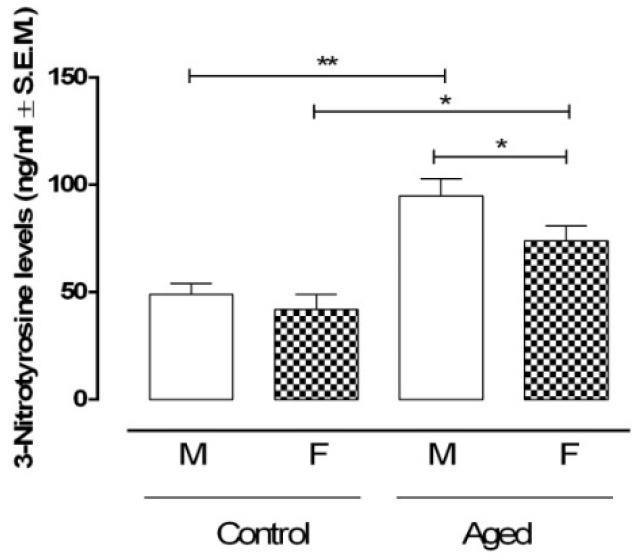
3-Nitrotyrosine retinal levels. ELISA determination of 3-Nitrotyrosine as a marker of peroxynitrite formation in aged and younger (control) retina. Data are expressed as mean ± S.E.M. of N = 10 retinas for each group. M: males; F: females; ** *p* < 0.01; * *p* < 0.05.

**Table 1 ijms-22-06280-t001:** Pearson’s r values evaluating the strength of association between RvD1 and retinal markers evaluated.

	Iba-1 (DU)	GFAP (DU)	NF-kB (µg/mL)	TNF-α (pg/mL)	Caspase 3 (ng/mL)	3-Nitrotyrosine (ng/mL)
RvD1 (IU/mL)	−0.88 **	−0.86 **	−0.93 **	−0.94 **	−0.82 **	−0.87 **

** *p* < 0.01.

## Data Availability

All data relevant to the study are included within the article.

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
