# Peer review of "Sex and Age-Related Differences in Neuroinflammation and Apoptosis in Balb/c Mice Retina Involve Resolvin D1"

_ijms, 2021, doi:10.3390/ijms22126280_

Round 1

Reviewer 1 Report

  1. change in Resolvin D1 level during aging is progressive process, this study only used two time points to examine the alteration of Resolvin D1 concentration during aging, 3 months and 24 months. There should be another intermediate time point sush as  12 month or 18 month of age to show prgressive change in Resolvin D1.
  2. Author used Balb/C mice to conclude the  gender differences of Resolvin D1 between male and female mice. It is not clear whether other mouse strains will generate similar results, so this conclusion is pre-mature. In fact, not all age-related  diseases are associated with gender.
  3. Immunohistochemistry results are not very convincing. Author should show lower magnificartion images with bigger region of staining to show overall staining results.
  4. Auhtor claimed that  n=10, is there any discperancy between animals  in RvD1 levels and other proteins examined?

Author Response

Reviewer 1

The authors thank this reviewer for his positive comments on the paper. They point out that an extensive revision of the text has been made due to the criticism of this reviewer and to the criticism raised by the Reviewer 2. Here are details of the answers.

  1. The authors agree with this reviewer that aging is a progressive process and a progressive monitoring of the ocular changes give clear picture of what’s happening, therefore they deleted any speculation that could misinterpret the results. For example, already in the introduction it was specified that the study is aimed at making a simple comparison between aged mice and younger mice (see Introduction last paragraph) as regards the morphological and biochemical characteristics of the retina in relation to the sex. The word “aging mice” has been substituted with with the word “aged mice” through the text, anyway.
  2. Taking into account that not all age-related diseases are associated with gender the authors replaced the word “gender” with the word “sex” more appropriate for the aim of the study. In addition, a sentence concerning the need to test other different mice strains in order to have conclusive data on the role of RvD1 in neurodegeneration has been added as last line of Discussion.
  3. Immunohistochemistries have been improved by adding lower magnification. I hope they are satisfactory now. However,the authors prefer to see the enlarged images in order to get cellular details. For example, for Nf-kB, which is a transcription factor, activation is evident only when we have colored nuclei (which certify nuclear translocation). At x4 objective this is not possible.

  4. Not at all, the legend misinterptrets the n number. It has now been conformed to the others.
  5. As a consequence of the major revision the references have been shortened.

Reviewer 2 Report

   In the study, Trotta et al evaluated the gender- and age-related neuroinflammation and apoptosis in eye. Especially, they focussed on the effect of the retinal levels of RvD1, a lipid mediator that they have studied the protection effects of this molecule on inflammation and degeneration of retina.  They found that the retinal RvD1 levels were decreased in aged mice when compared to younger mice, and that the decrease is markedly larger in males. They also found that several aspects of retinal impairment (the  astrocyte activation, neuroinflammation, apoptosis and nitrosative stress, in line with the microglia and Müller cell hypertrophy) positively correlated to the retinal levels of RvD1 in aged mice. From these data, they concluded that RvD1 endogenous levels seem to be associated to neuroprotection in aged female retinas.   While each of their data seems to be thought-provoking, it cannot be denied that the lack of logic makes the claims of this manuscript inappropriate.   The problem is that, while there are clear correlations between RvD1 levels and other retinal defects, it is not always clear whether there are causal relationships between the data. The author's previous works on RvD1 has focused on how RvD1 in the eye or in cultured cells prevents various types of retinal dysfunction caused by drug administration (ref 9-12). In these kind of studies, the causal effects of RvD1 on various changes are clear. On the other hand, when changes in RvD1 abundance and retinal degeneration are measured as a function of aging, as in the present study, it is unclear whether there is a causal relationship, even if a high correlation parameters are calculated. In general, it needs further investigations to judge whether the high correlation is derived from causal relationship or not. So far, the protection effects of RvD1 has been extensively studied by authors, and thus, this reviewer also think it is highly possible that the higher RvD1 level in retina results in anti-aging effects. However, logically, we cannot rule out the possibility that causality is only apparent.    For example, RvD1 and some retinal degenerative changes may not have a direct causal relationship, although they have a common cause associated with aging. In order to clearly state the protective effect of RvD1 in age-related retinal disfunctions, it seems to be essential to show that the administration of RvD1 to retina improves the changes associated with aging.  Asserting only the author's hypothesis without conducting such an experiment clearly lacks logic and is a serious flaw in a scientific paper. Therefore, this reviewer believe that it is imperative to conduct a new experiment or to logically restructure the paper.   The minor concerns (7 points) 1: line 41, conclusions:  As pointed out in “Major concern”, the statement of “association” is not clear.    2: photos in figures    In figure 2, 3, 4, 5, and 6, photos of retina are shown. The retina-specific structure are well indicated but there are no explanations of what is what. What is the each layer? It would be better to give a little more detailed explanation for readers in a wider field.   3: The retinal structure in figure 3.   In figure 3, the retinal structures of aged mice are highly disordered when compared to aged retinae shown in other figures.   If possible, please explain the difference.   4: figure legend of figure 3 (line 130)     “d. aged male” seems to be “d.aged female”.   5: insets in figure 6   In the panels c and d of Fig. 6A, the retinal structures seems to be displayed in an expanded scale as insets. However, the information of these inset is not given.    6: discussion line 265-266 It is written that “some studies showed a higher incidence and severity of late-stage AMD in women”.  It looks that this description goes against the results of this experiment. Are 24-week old female mice in conditions of postmenopausal, or not?    7:”relationship” in 277 In line 277, it is written that “its relation”. It cannot be determined whether this relationship is a causal relationship or not. As pointed out in “major concern”, please state this clearly.

Author Response

Reviewer 2

This reviewer has raised positive comments for the paper which according to his suggestions has been substantially revised. Thanks.

Considering all the logical criticisms that the reviewer 2 made on this paper, the authors decided to delete from the paper any personal opinion or speculation beyond the pure observational data. They thank the reviewer and delete all the text that was not supported by a causal relationship regarding RvD1 levels and retinal degeneration / neuroprotection. Consequently, i) only the description of what was found in the retinas of aged mice compared to the retinas of younger mice was left; ii) the hasty conclusion that RvD1 is a neuroprotector in this setting has been rewritten; iii) the speculation made about the hormonal mechanism that would have led to differences in RvD1 levels between males and females, and the consequent differences in the monitored markers, has been eliminated. The authors left the correlation data between the observed events hoping that they would not lead to misinterpretation among readers and reserve the right to conduct experiments aimed at ascertaining the cause-and-effect relationship for RvD1 in a context based on repeated administration of precursors of RvD1 in aged mice. For example, administration of docosohexaenoic acid DHA.

References have been shortened.

As per reviewer suggestion the authors modified some minor points also. 

  1. The statement “association” in the conclusion is cancelled since the conclusion has been changed.
  2. Figures 2,3,4,5,6. Immunohistochemistries have been improved by adding lower magnification. I hope they are satisfactory now. However,the authors prefer to see the enlarged images in order to get cellular details. For example, for Nf-kB, which is a transcription factor, activation is evident only when we have colored nuclei (which certify nuclear translocation). At x4 objective this is not possible.

The pictures are now explained and we changed also the legends.

  1. Regarding the retinal structure of aged mice in Figure 3 (panel A), it is likely that the reported appearance results from the technique. We performed the epitope retrieval at 95 ° C for a longer time than Iba-1. We already mentioned in the section 4.3 of materials and methods. Electron microscopy of Figure 3C has been detailed in results (section 2.3, last lines).
  2. Figure 3 legend is correct now.
  3. Figure 6 has been changed.
  4. The text concerning this point has been deleted.
  5. The speculation concerning the causal relationship is cancelled with a new conclusion rewritten.

Round 2

Reviewer 2 Report

In the revised version, the major problem with the logical aspect that I pointed out have been greatly improved. I believe it can be accepted as is.

Author Response

The authors thank the Reviewer for acceptance.